# Difference in pyruvic acid metabolism between neonatal and adult mouse lungs exposed to hyperoxia

**Kosuke Tanaka, Takaaki Watanabe, Junichi Ozawa, Masato Ito, Nobuhiko Nagano, Yukio Arai, Fuyu Miyake, Shun Matsumura, Shingo Kobayashi, Ryuta Itakura, Fumihiko Namba** *

Department of Pediatrics, Saitama Medical Center, Saitama Medical University, Kawagoe, Saitama, Japan

* nambaf@saitama-med.ac.jp

**Data Availability Statement:** All relevant data are within the manuscript and its Supporting Information files.

## Abstract

### Objective

Neonatal lungs are more tolerant to hyperoxic injury than are adult lungs. This study investigated differences in the response to hyperoxic exposure between neonatal and adult mouse lungs using metabolomics analysis with capillary electrophoresis time-of-flight mass spectrometry (CE- TOFMS).

### Methods

Neonatal and adult mice were exposed to 21% or 95% $O_2$ for four days. Subsequently, lung tissue samples were collected and analyzed by CE-TOFMS. Pyruvate dehydrogenase (PDH) enzyme activity was determined using a microplate assay kit. PDH kinase (*Pdk*) 1, *Pdk2*, *Pdk3*, and *Pdk4* mRNA expression levels were determined using quantitative reverse transcription-polymerase chain reaction. Pdk4 protein expression was quantified by Western blotting and Pdk4 protein localization was evaluated by immunohistochemistry.

### Results

Levels of 3-phosphoglyceric acid, 2-phosphoglyceric acid, phosphoenolpyruvic acid, and lactic acid were significantly elevated in the lungs of hyperoxia-exposed versus normoxia-exposed adult mice, whereas no significant differences were observed with hyperoxia exposure in neonatal mice. PDH activity was reduced in the lungs of adult mice only. *Pdk4* mRNA expression levels after hyperoxic exposure were significantly elevated in adult mice compared with that in neonatal mice. Conversely, gene expression levels of *Pdk1*, *Pdk2*, and *Pdk3* did not differ after hyperoxic exposure in either neonatal or adult mice. Pdk4 protein levels were also significantly increased in adult mouse lungs exposed to hyperoxia and were localized mainly to the epithelium of terminal bronchiole.

**Funding:** This research was supported by the Saitama Medical University Internal Research Grant (Grant no. 17-C-1-2) (SK) and Saitama Medical University Young Doctors Educational Grant (Grant no. 29-F-1-01) (RI) from Saitama Medical University, a grant from Kawano Masanori Memorial Public Interest Incorporated Foundation for Promotion of Pediatrics (Grant no. 29-7) (FN), and Grants-in-Aid for Scientific Research (KAKENHI) from the Japan Society for the Promotion of Science (Grant no. 18K15729) (SK). The funders had no role in study design, data collection and analysis, decision to publish, or preparation of the manuscript.

**Competing interests:** The authors have declared that no competing interests exist.

## Conclusions

Specific metabolites associated with glycolysis and gluconeogenesis were altered after hyperoxia exposure in the lungs of adult mice, but not in neonates, which was likely a result of reduced PDH activity due to *Pdk4* mRNA upregulation under hyperoxia.

## Introduction

Exposure to high oxygen ($O_2$) level induces lung injury in newborn rodents, which are often used as an animal model of neonatal chronic lung disease due to its pathological similarity to bronchopulmonary dysplasia that occurs in humans [1]. Furthermore, adult mice exposed to hyperoxia in the neonatal period exhibit persistent alveolar simplification, increased lung compliance, increased sensitivity to viral infection, and pulmonary vascular disease [2, 3]. Hyperoxia also causes acute lung injury and acute respiratory distress syndrome in adult mice and humans [4]. Although most neonatal rodents survive hyperoxia-induced lung injury, adults die within 1 week of exposure to high $O_2$ concentrations [5]. Elevated antioxidant enzyme activity [6] and decreased superoxide-generating capacity [7] have been proposed as underlying mechanisms for the enhanced tolerance to hyperoxia in newborn animals, the exact cause is yet to be clarified.

Pyruvate dehydrogenase (PDH) catalyzes the oxidative decarboxylation of pyruvate to produce acetyl-coenzyme A (acetyl-CoA). PDH activity is negatively regulated by PDH kinase (PDK). There are four known tissue-specific PDK isoenzymes: PDK1and PDK 2, which are expressed in most cells; PDK3, which is expressed mainly in the testes: and PDK4, which is expressed primarily in muscle during metabolic stress [8]. PDK4 is located in the mitochondrial matrix and regulates the rate of glucose oxidation and fatty acid metabolism in mammalian cells through direct inhibition of the PDH complex. In 1980's, Kimura, *et al*. reported that total PDH activity was 45% lower in the lungs of neonatal rats exposed to 100% $O_2$ for the first 6 days of life than that in normoxic controls [9]. However, the role of pyruvate metabolism differences between newborn and adult mice in the effects of hyperoxic exposure has not been investigated.

Metabolites, end products of cellular regulatory processes, play important biological roles and provide a functional readout of cellular biochemistry that is important to identifying metabolic pathways related to disease processes. Metabolomics is an analytical technique in systems biology used to characterize low-molecular-weight metabolites in biological sample such as cells, tissues, and biofluids under a particular condition. This technique has been widely used to identify novel biomarkers and gain insight into the mechanisms that underlie various diseases [10, 11]. Few metabolomic analyses were performed on mouse lungs exposed to hyperoxia, and none have been conducted to elucidate differences in susceptibility to high concentrations of $O_2$ between neonatal and adult mice. Therefore, in this study, we aimed to investigate differences in the response to hyperoxic exposure between neonatal and adult mouse lungs using metabolomic analysis with capillary electrophoresis time-of-flight mass spectrometry (CE-TOFMS).

## Materials and methods

### Animals

All procedures and protocols were approved by the Animal Care and Use Committee of Saitama Medical University (Permit no. 1691). C57BL/6J mice (SLC, Hamamatsu, Shizuoka,

Japan) were maintained on a 12-h light, 12-h dark cycle with ad libitum access to food and water. All experimental procedures were performed in accordance with the National Institutes of Health Guidelines for Laboratory Animal Care, in addition to our institutional guidelines. Neonatal pups (<12 hours old) and adult mice (8 weeks old) were randomly assigned to normoxia (room air) or hyperoxia (95% $O_2$). Details of the hyperoxic exposure and lung tissue collection were described in our previous report [12]. Briefly, mice were exposed to hyperoxia for 96 h in a chamber. Mice were anesthetized with an intraperitoneal injection of pentobarbital (50 mg/kg), and the right lung was excised, snap-frozen with liquid nitrogen, and stored at −80˚C for further analysis. All efforts were undertaken to minimize suffering.

## Metabolite extraction

Metabolite extraction and metabolomic analysis were conducted at Human Metabolome Technologies (HMT) (Tsuruoka, Yamagata, Japan). Neonatal mice of unknown sex (<12 h old, n = 8) and adult male and female mice (8 weeks old, n = 4 each) were exposed to normoxia or hyperoxia for 96 h. Approximately 50 mg of frozen lung tissue from four animals in each group was immersed into 1500 μL of 50% acetonitrile/Milli-Q water containing internal standards (Solution ID: 304–1002, HMT) at 0 ˚C to inactivate enzymes. The tissue was homogenized five times at 1500 rotation per minutes for 120 s using a tissue homogenizer (Micro Smash MS100R, Tomy Digital Biology Co., Ltd., Tokyo, Japan), and the homogenate was centrifuged at 2300*g* at 4˚C for 5 min. Subsequently, 800 μL of supernatant was filtered by centrifugation through a Millipore 5-kDa cutoff filter at 9100*g* at 4˚C for 120 min to remove proteins. The filtrate was concentrated by centrifugation and resuspended in 50 μL Milli-Q water for CE-TOFMS analysis at HMT.

## Metabolomic analysis

Metabolomic analysis was conducted by the BasicScan package by HMT using CE-TOFMS based on previously described methods (9, 10). Briefly, CE-TOFMS analysis was performed using an Agilent CE system equipped with an Agilent 6210 TOFMS, Agilent 1100 isocratic high-performance liquid chromatography HPLC pump, Agilent G1603A CE-MS adapter kit, and Agilent G1607A CE-ESI-MS sprayer kit (Agilent Technologies, Waldbronn, Germany). The systems were controlled by Agilent G2201AA ChemStation software version B.03.01 for CE (Agilent Technologies) and connected by a fused silica capillary (50 μm *i.d.* × 80 cm total length) with commercial electrophoresis buffer (H3301-1001 and H3302-1021 for cation and anion analyses, respectively, HMT) as the electrolyte. The spectrometer was scanned from m/z 50–1000 (9). Peaks were extracted using MasterHands, an automatic integration software (Keio University, Tsuruoka, Yamagata, Japan), to obtain peak information including m/z, peak area, and migration time (11). Peaks were annotated according to the HMT metabolite database based on m/z values with migration times. Areas of the annotated peaks were then normalized on the basis of internal standard levels and sample amounts to obtain relative level for each metabolite. In annotation, mass error was limited within ±10ppm of theoretical value and migration tome was limited within ±0.5 minutes of the database.

## Pyruvate dehydrogenase enzyme activity assay

PDH enzymatic activity was measured using a commercially available PDH enzyme activity microplate assay kit (ab109902; Abcam, Cambridge, UK) following the manufacturer's instructions. Briefly, lung tissue extracts (800 μg) from neonatal mice (<12 h old, n = 24) and adult male and female mice (8 weeks old, n = 12 each) exposed to normoxia or hyperoxia for 96 h were loaded into a 96-well plate for PDH activity assay. The PDH enzyme complex

components were immunocaptured by monoclonal antibodies coated on the wall of each well, and PDH enzymatic activity was determined on the basis of the production of reduced nicotinamide adenine dinucleotide coupled to the reduction of the reporter dye WST-1 to the color yellow. $V_{max}$ and absorbance of the reporter dye in each well were measured at 450 nm at room temperature using a kinetic program for 30 min (Corona Grating Microplate Reader SH-9000, Corona Electric, Ibaraki, Japan).

### Gene expression analysis

Total RNA was extracted from excised lungs of neonatal mice (<12 hours old, n = 12) and adult male and female mice (8 weeks old, n = 6, each) exposed to normoxia or hyperoxia for 96 h as previously described [13], and mRNA expression levels of *Pdk1* (Assay ID: Mm00554300_m1), *Pdk 2* (Assay ID: Mm00446681_m1), *Pdk 3* (Assay ID: Mm00455220_m1), and *Pdk 4* (Assay ID: Mm01166879_m1; Applied Biosystems, Tokyo, Japan) were evaluated by polymerase chain reaction (PCR) using the TaqMan® fluorogenic detection system. Because hyperoxia alters four of the housekeeping genes recommended by the manufacturer (*Hprt1*, *Hsp90ab1*, *Gapdh*, and *Actb*), we used β-glucuronidase (Mm01197698_m1) for normalization after confirming that its expression levels were comparable among all conditions.

### Western blot

Western blot analyses were performed to evaluate PDK4 protein levels. Lung tissues excised from neonatal mice (<12 h old, n = 12) and adult male mice (8 weeks old, n = 12) exposed to normoxia or hyperoxia for 96 h were homogenized in tissue extraction buffer (78510; Thermo Fisher Scientific, Waltham, MA, USA). The lysates were clarified by centrifugation at 10,000 g for 5 min. Protein samples (50 μg) were separated using SDS/PAGE and transferred to membranes. The membranes were blocked and incubated with the anti-PDK4 primary antibody (1:500; ab172920; Abcam) or anti-β-actin antibody (1:5000; GTX26276; GeneTex, Irvine, CA, USA), followed by horseradish peroxidase (HRP)-labeled secondary antibodies. Chemiluminescence was detected using the Amersham ECL Prime Western blotting detection reagent (GE Healthcare, Chicago, IL, USA) with a digital imaging system (Bio-Rad ChemiDoc XRS+; Bio-Rad Laboratories, Hercules, CA, USA).

### Immunohistochemistry

Formalin-fixed paraffin-embedded sections of lungs excised from neonatal mice (<12 h old, n = 12) and adult male mice (8 weeks old, n = 12) exposed to normoxia or hyperoxia for 96 h were deparaffinized and rehydrated. The sections were processed with Antigen Unmasking Solution pH6 (H-3300; Dako, Glostrup, Denmark) and immersed in 3% hydrogen peroxide solution for 30 min. The sections were incubated with an anti-PDK4 primary antibody (1:200; PDK4(12949-1-AP); ProteinTech, Rosemont, IL, USA) for 1 h at room temperature. Immunoreactivity was detected using the Envision Kit with a HRP-labelled anti-rabbit IgG secondary antibody (K4003; Dako). Sections were counterstained with Mayer's hematoxylin.

### Statistical analysis

Group comparisons were performed using Welch's *t* test. Principal component analysis (PCA) and hierarchical cluster analysis (HCA) were performed using two proprietary software packages by HMT called SampleStat and PeakStat, respectively. Detected metabolites were plotted on metabolic pathway maps using the VANTED software [14].

Analyses of PDH activity, gene expression, and protein expression were conducted using the R statistical package (version 3.5.2), and values were represented as means ± standard error of the mean (SEM). The null hypothesis was tested by unpaired Student's *t* test for comparisons between two treatment groups. A p value < 0.05 was considered statistically significant.

## Results

### Effects of hyperoxic exposure on metabolites in the newborn and adult mouse lungs

Metabolites extracted from newborn and adult mouse lungs exposed to normoxia or hyperoxia were identified using metabolomics. There were 258 peaks identified and quantified with metabolite standards by matching closest m/z values and normalized migration times. Of these, 130 and 128 were cations and anions, respectively. The levels of 63 and 112 metabolites in the newborn and adult mouse lungs, respectively, were significantly altered by hyperoxic exposure.

The levels of 114 metabolites were significantly differed between the newborn and adult mouse lungs under normoxic condition, whereas the levels of 99 metabolites were significantly different between the newborn and adult mouse lungs after hyperoxic exposure. Four major clusters were distinguished by the HCA (Fig 1), and these clusters were coincident with the four groups from which the lungs were obtained: room air newborn, high $O_2$ concentration newborn, room air adult, and high $O_2$ concentration adult groups. The PCA revealed that the metabolic profiles of the newborn and adult mouse lungs were clearly distinct from each other (first principal component), as were the metabolic profiles of the mouse lungs exposed to hyperoxia and normoxia (second principal component; Fig 2). The cumulative contribution rate of these two major principal components was 59.5%.

### Metabolic pathways altered in hyperoxia-induced lungs differ between adults and neonates

The peaks detected by CE-TOFMS were classified into glycolysis/gluconeogenesis, pentose phosphate pathway, tricarboxylic acid (TCA) cycle, urea cycle, purine/pyrimidine metabolism, coenzyme metabolism, and various amino acid metabolic pathways based on the candidate compounds (S1 Fig). On the basis of the KEGG pathway database, metabolic pathways and functional classifications were constructed. Among the numerous pathways identified, we focused on the glycolysis-citric acid circuit because many metabolites constituting this circuit were highly loaded in both the first and second principal components in the PCA (S1 Table).

Higher levels of 3- phosphoglyceric acid (PG), 2-PG, phosphoenolpyruvate (PEP), and lactic acid, which are involved in the glycolysis and gluconeogenesis pathways were found in the hyperoxia-exposed adult lungs than in the hyperoxia-exposed neonatal lungs. Hyperoxia increased the levels of these metabolites in the adult mouse lungs, whereas hyperoxia had no significant effect on their levels in neonatal mouse lungs (Fig 3).

### Reduced PDH enzymatic activity in the hyperoxia-exposed adult mouse lungs

On the basis of the metabolomics analysis, we speculated that the differences in the glycolysis-citric acid metabolic pathway between the newborn and adult mouse lungs exposed to hyperoxia stemmed from differences in the conversion of pyruvic acid to acetyl-CoA. As this

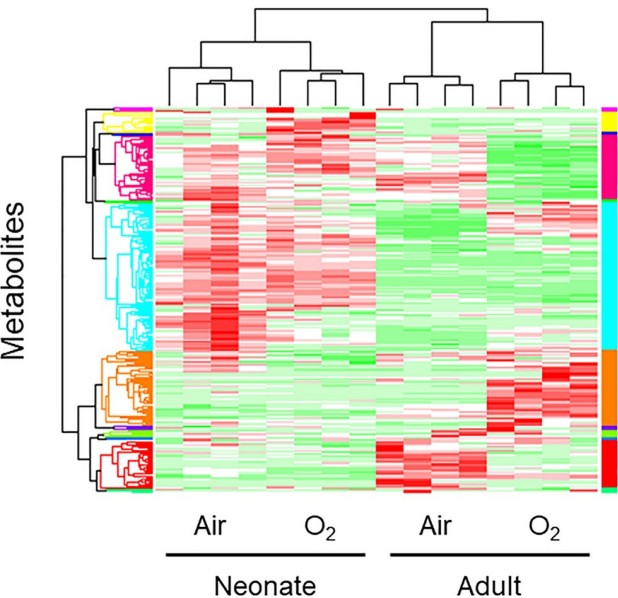

**Fig 1. Visualization of the profiles of the detected metabolites uasing a heat map.** Columns represent individual mice, and row represent specific metabolites. The color scale from green to red indicates relative changes in the levels of the detected metabokites. Red indicates that the relative level of the metabolite is higher than the average, and green indicates that the relative level of the metabolite is lower than the average. Lungs were obtained from four different groups of mice: room air newborn, high $O_2$ concentration newborn, room air adult, and high $O_2$ concentration adult groups. The four major clusters distinguished by HCA are coincident with the four groups.

oxidative decarboxylation reaction is facilitated by PDH, we quantified its enzymatic activity (Fig 4) and determined that it was significantly reduced in the hyperoxia-exposed than in the normoxia-exposed adult mouse lungs. However, a similar reduction in PDH enzymatic activity due to hyperoxia exposure was not detected in the newborn lungs.

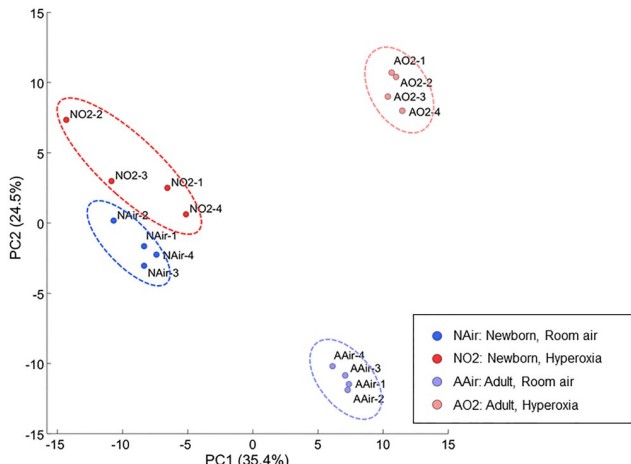

**Fig 2. Principal component analysis of the metabolomic dataset of the four groups.** Percentage on the axes represent the contribution rate of the first (PC1) and second (PC2) principal components. PC1 and PC2 had contribution ratios of 35.4% and 24.5%, respectively, and were sufficient to distinguish the four groups. Ellipses are graphical representations of groups without any statistical support. NAir, newborn, room air; NO2, newborn hyperoxia; AAir, adult room air; AO2, adult, hyperoxia; PC, principal component.

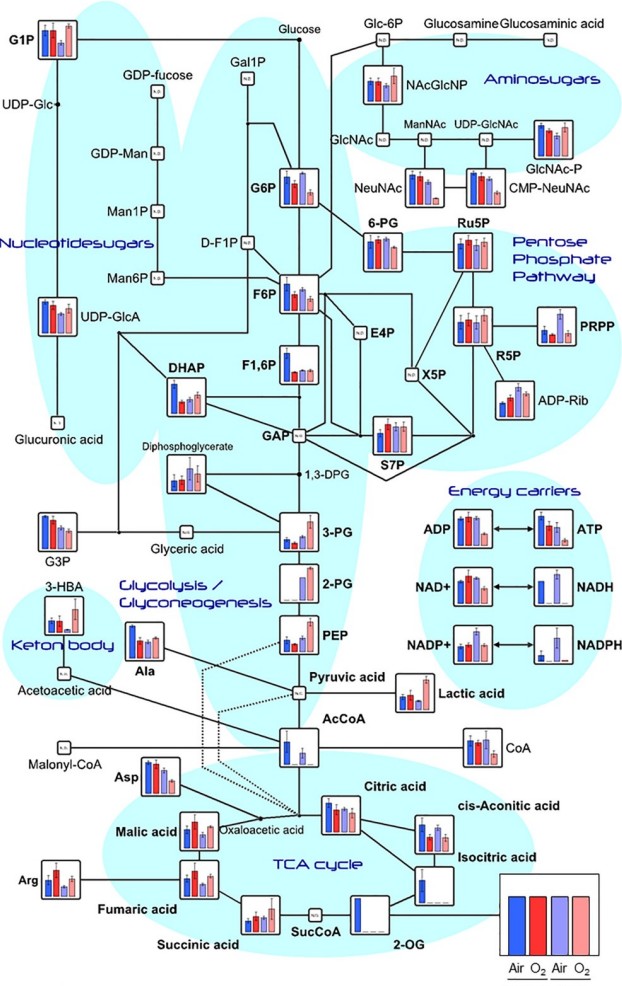

**Fig 3. Metabolomic data map of the glycolysis and glyconeogenesis pathways in the neonatal and adult mouse lungs.** Vertical axes show the detected average value with standard deviation (error bar) for each comparison, and the groups are arranged from the left in the following order: room air newborn (blue), high $O_2$ concentration newborn (bright red), room air adult (purple), and high $O_2$ concentration adult (light red). All metabolic data are presented as means ± standard deviation from biological triplicate samples.

## Differences in *Pdk4* mRNA and protein expression levels in hyperoxia-exposed lungs

PDK negatively regulates PDH that catalyzes the oxidative decarboxylation of pyruvate to produce acetyl CoA. Therefore, we quantified the mRNA expression levels of *Pdk1*, *Pdk2*, *Pdk3*, and *Pdk4* by real-time PCR in the neonatal and adult mouse lungs exposed to either normoxia or hyperoxia (Fig 5). Lung expression of *Pdk4* was higher in adult mice than in neonates under normoxia, and expression of *Pdk4* was increased after exposure to hyperoxia in both groups. Additionally, the expression of *Pdk4* was lower in hyperoxia-exposed neonate lungs than in normoxia-exposed adult lungs. In contrast, no significant changes were observed in the lung expression levels of *Pdk1*, *Pdk2*, or *Pdk3* in newborn or adult mice after hyperoxic exposure. Western blotting confirmed the increase Pdk4 protein levels in the adult mouse lungs exposed to hyperoxia (Fig 6A and 6B). We also evaluated the expression and localization of Pdk4

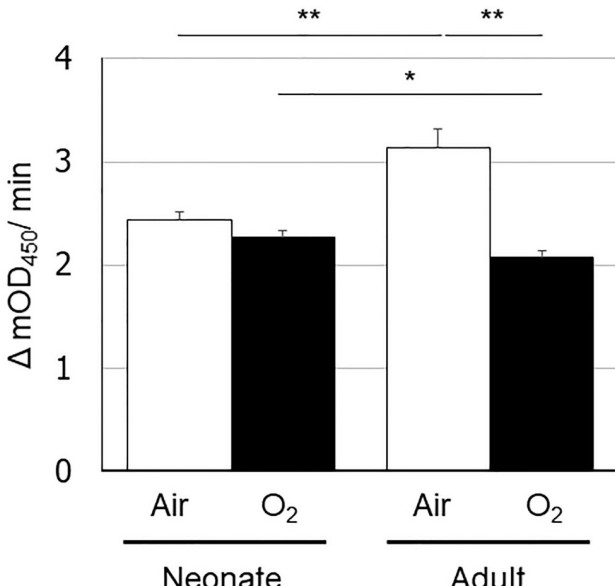

**Fig 4. Pyruvate dehydrogenase enzymatic activity levels in the neonatal and adult mouse lungs exposed to normoxia or hyperoxia.** The vertical axis shows PDH enzymatic activity ($\Delta$ mOD$_{450}$/min). Animals were exposed to normoxia (open bars) or hyperoxia (filled bars). Data are shown as means ± SEM. Comparisons between groups were performed using a *t* test. * p < 0.05; ** p < 0.01. Air: normoxia; O2: hyperoxia; SEM: Standard error of the mean.

protein in the lungs by immunohistochemistry. Immunohistochemical staining was observed mainly in the epithelium of terminal bronchiole in all four groups (Fig 6C).

## Discussion

This is the first study to identify differences in the metabolic profiles of newborn and adult mouse lungs after hyperoxic exposure using metabolomics analysis. Our analyses revealed

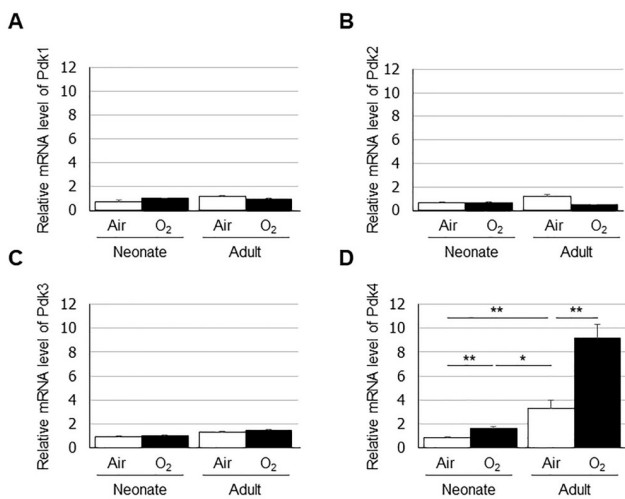

**Fig 5. Expression levels of (A) *Pdk1*, (B) *Pdk2*, (C) *Pdk3*, and (D) *Pdk4* in the lungs quantified by real-time PCR.** Data are shown as means ± SEM. Comparisons between groups were performed using a *t* test. * p < 0.05; ** p < 0.01. Air: normoxia; O2: hyperoxia; SEM: Standard error of the mean.

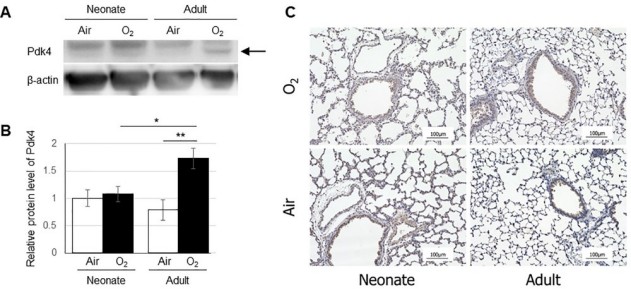

**Fig 6. Expression levels of Pdk4 protein in the lungs quantified by Western blot (A,B) and immunohistochemical localization of Pdk4 protein in the lung (C).** (A) The arrow indicates a band of Pdk4 protein. (B) The vertical axis shows the relative expression of Pdk4 protein. Data are shown as means ± SEM. Comparisons between groups were performed using a $t$ test. $^*$ $p < 0.05$; $^{**}$ $p < 0.01$. Air: normoxia; O2: hyperoxia; SEM: Standard error of the mean. (C) Immunostaining was observed in the epithelium of terminal bronchiole in all four groups (brown). Scale bar = 100 μm.

major differences between newborn and adult mice in metabolites constituting the glycolysis-citric acid circuit, including 3-PG, 2-PG, PEP, and lactic acid. We also found that the lung mRNA expression and protein levels of *Pdk4* were upregulated after hyperoxic exposure in adult mice, which led to reduced PDH activity in the lung; this was not observed in neonates. These findings partially explain the differences in susceptibility to hyperoxia between adults and neonates.

Several studies have investigated the differences in O2 tolerance between newborn and adult animals. Ischiropoulos, *et al.* studied neonatal and adult rats exposed to hyperoxia and reported that O2 tolerance in neonates can be explained in part by a favorable balance between antioxidant defenses and subcellular superoxide-generating capacity [7]. Another group observed the ability of the neonatal type II cell to respond to hyperoxia with an early increase in antioxidant enzyme activity of superoxide dismutase, which may contribute to the enhanced O2 tolerance of the neonatal rat [15]. Furthermore, compared with adult rodents, newborn rodents reportedly have a diminished pulmonary inflammatory response due to differences in chemokine levels [16, 17]. L-selectin on neutrophils was lower in neonatal rats than in adult rats, and soluble L-selectin was higher in the air-exposed neonate and tended to be higher after hyperoxic exposure. This suggested that both lower surface L-selectin and higher soluble L-selectin might play a role in the diminished pulmonary inflammatory response in the neonatal rat after hyperoxia [17]. However, there no studies have focused on metabolic changes in lungs exposed to hyperoxia using a metabolomics approach to elucidate the differences in O2 tolerance between newborns and adults.

The metabolomics analysis in adult mice revealed increases in the levels of glycolysis pathway metabolites, with a decrease in acetyl CoA levels. Glycolysis involves 10 reactions and converts glucose into pyruvic acid, producing four molecules of adenosine triphosphate and two molecules of reduced nicotinamide adenine dinucleotide [18]. In the last steps of glycolysis, 3-PG is converted into pyruvate via the synthesis of 2-PG and PEP, with the concomitant conversion of adenosine diphosphate into adenosine triphosphate. Subsequently, pyruvate is converted into acetyl CoA by PDH, leading to an increased influx of acetyl CoA into the TCA cycle. As oxidative phosphorylation in the mitochondrial electron transport chain and glycolysis are the energy-yielding pathways in cells, previous studies have focused on the effects of hyperoxia on glycolysis and mitochondrial metabolism in adults [19–27].

Although the effect of hyperoxia on glycolysis is not well understood, numerous studies have demonstrated an increase in the rate of glycolysis during hyperoxia [19, 25, 26]. Simon *et al.* showed that the activity levels of important glycolytic pathway enzymes such as

glyceraldehyde-3-phosphate dehydrogenase, pyruvate kinase, and phosphofructokinase did not decrease in hyperoxia [26]; these finding are consistent with the results of our metabolomic analysis. The effect of hyperoxia on mitochondrial metabolism, albeit investigated previously [20–23], is also not well understood. Bassett *et al*. assessed mitochondrial activity and changes in glucose metabolism in the perfused rat lung after hyperoxic exposure. They reported lower maximal rates of pyruvate catabolism and increased lactic acid formation upon perfusion with glucose after hyperoxic exposure. These results demonstrated that mitochondrial metabolism of pyruvate, rather than glycolysis, is potentially the initial step in $O_2$-induced inhibition of energy metabolism [20, 21] and are consistent with the findings of our metabolomic analysis. Bassett *et al*. also examined the effect of *in vivo* $O_2$ exposure on the lung mitochondrial energy-generating pathways and demonstrated that, although baseline metabolic rates were sustained during the first 24–30 h of exposure, maximal pyruvate and palmitate carbon dioxide production was compromised. The authors concluded that, in addition to the PDH complex, nicotinamide adenine dinucleotide-linked isocitrate dehydrogenase was a potential site of $O_2$-induced mitochondrial damage [22]. However, our analysis did not reveal similar findings. In addition to alterations in the TCA cycle, Kumuda *et al*. showed that hyperoxia caused dysfunction of both mitochondrial complexes I and II and disrupted mitochondrial electron transport chain function [23].

In the metabolomic analysis, we identified differences in the levels of metabolites that constitute the glycolysis-citric acid circuit, including 3-PG, 2-PG, PEP, and lactic acid, between the newborn and adult mouse lungs after hyperoxic exposure. Our analysis demonstrated increased levels of metabolites upstream of the reaction that converts pyruvate to acetyl-CoA in the hyperoxia-exposed adult mouse lungs. These differences were not observed in the neonatal mouse lungs.

We hypothesized that the increased levels of these metabolites in the adult mouse lungs were due to reduced PDH activity and confirmed this in the adult mouse lung exposed to hyperoxia. This effect was not observed in the newborn lungs exposed to hyperoxia when compared with their normoxic counterparts. This suggests that hyperoxia perturbed metabolic pathways and induced increased glycolysis in the adult mouse lungs, which is in agreement with the findings of a previous study [28].

In the current study, we quantified the mRNA expression levels of the four PDK isoenzymes and found that the lung expression levels of *Pdk4* were significantly increased in both the neonates and the adults after hyperoxic exposure. However, compared with those in the neonatal mice, remarkable increases in *Pdk4* expression were observed in the adult mouse lungs after hyperoxic exposure. This suggests that increased *Pdk4* negatively modulated PDH enzymatic activity, resulting in the accumulation of 3-PG, 2-PG, PEP, and lactic acid in the adult mouse lungs exposed to hyperoxia.

Lingappan *et al*. compared sex-specific differences in the pathophysiology of hyperoxic lung injury [29] based on observed sex-specific differences in susceptibility to hyperoxic lung injury in both animal models and epidemiological studies of human patients. Males with acute respiratory distress syndrome have a higher mortality rate than have females, and male sex is an independent risk factor for the development of bronchopulmonary dysplasia in premature neonates. The authors performed microarray analysis of the lungs after hyperoxia exposure and demonstrated that the *Pdk4* gene was upregulated in males. This result is consistent with our findings suggesting that increased *PDK4* expression might confer susceptibility to hyperoxia.

We speculate that the reduced PDH activity in the adult mouse lungs due to elevated expression of *Pdk4* might lead to a decrease in the rate of transition from glycolysis to the TCA cycle, resulting in reduced energy production via oxidative phosphorylation. This limitation of

energy supply to the lung cells might be one of the underlying reasons for vulnerability to hyperoxia in adult mice.

In conclusion, we demonstrated that specific metabolites related to glycolysis and gluconeogenesis, including 3-PG, 2-PG, PEP, and lactic acid, were increased in the adult but not neonatal mouse lungs after hyperoxic exposure. These increased metabolite levels in the adult mouse lungs might be due to reduced PDH activity, which was negatively regulated by *Pdk4* upregulation under hyperoxia. These result provide a possible explanation for the difference in tolerance to hyperoxia between neonatal and adult mice and an insight into potential intervention approaches for preventing hyperoxia-induced acute lung injury and acute respiratory distress syndrome.

## Supporting information

**S1 Fig. Metabolomic data map of the all metabolic pathway detected in the neonatal and adult mouse lungs.** For all metabolite graphs, the vertical axis shows the detected average value with standard deviation (error bar) for each comparative group, and the groups are arranged from the left in the following order: room air newborn (blue), high $O_2$ concentration newborn (bright red), room air adult (purple), and high $O_2$ concentration adult groups (light red). All metabolic data are presented as means ± standard deviation from biological triplicate samples.
(PPTX)

**S2 Fig.**
(JPG)

**S3 Fig.**
(JPG)

**S1 Table. Factor loadings of the principal component analysis (PCA).** Metabolites constituting glycolysis-citric acid circuit are highlighted with grey background.
(XLSX)

## Acknowledgments

The authors thank Dr. Shigeo Tojo and Human Metabolome Technologies for the metabolomic analysis and technical advices and Ms. Kikumi Matsuoka (Departments of Biomedical Sciences, Saitama Medical Center, Saitama Medical University) for technical assistance with the experiments.

## Author Contributions

**Conceptualization:** Fumihiko Namba.

**Data curation:** Kosuke Tanaka.

**Formal analysis:** Kosuke Tanaka.

**Funding acquisition:** Shingo Kobayashi, Ryuta Itakura, Fumihiko Namba.

**Investigation:** Kosuke Tanaka, Takaaki Watanabe, Junichi Ozawa, Masato Ito, Nobuhiko Nagano, Yukio Arai, Fuyu Miyake, Shun Matsumura, Fumihiko Namba.

**Methodology:** Fumihiko Namba.

**Supervision:** Fumihiko Namba.

**Validation:** Fumihiko Namba.

**Writing – original draft:** Kosuke Tanaka.

**Writing – review & editing:** Kosuke Tanaka, Takaaki Watanabe, Junichi Ozawa, Masato Ito, Nobuhiko Nagano, Yukio Arai, Fuyu Miyake, Shun Matsumura, Shingo Kobayashi, Ryuta Itakura, Fumihiko Namba.

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
