## [Decision Letter · Decision Letter 0]

29 Jul 2020

PONE-D-20-17370

Difference in pyruvic acid metabolism between neonatal and adult mouse lungs exposed to hyperoxia

PLOS ONE

Dear Dr. Fumihiko Namba,

Thank you for submitting your manuscript to PLOS ONE. After careful consideration, we feel that it has merit but does not fully meet PLOS ONE’s publication criteria as it currently stands. Therefore, we invite you to submit a revised version of the manuscript that addresses the points raised during the review process.

Though it is a revised submission, it has been assigned to new editor with a new manuscript number due to the fact that the revision has not been submitted on time. I could see reviewer comments and your revised version of the manuscript with all your modifications and responses. For a strong third opinion, I have sent it to another reviewer, who had revised it and asked for minor revision. 

I would be thankful to you if you could revise the manuscript and submit it for consideration. 

We look forward to receiving your revised manuscript.

Kind regards,

Ch Ratnasekhar, Ph.D.

Academic Editor

PLOS ONE

Journal Requirements:

2. Please ensure you have thoroughly discussed any potential limitations of this study within the Discussion section, including the limited sample size.

Reviewers' comments:

Reviewer's Responses to Questions

**Comments to the Author**

1. Is the manuscript technically sound, and do the data support the conclusions?

Reviewer #1: Yes

2. Has the statistical analysis been performed appropriately and rigorously? 

Reviewer #1: Yes

3. Have the authors made all data underlying the findings in their manuscript fully available?

Reviewer #1: Yes

4. Is the manuscript presented in an intelligible fashion and written in standard English?

Reviewer #1: Yes

5. Review Comments to the Author

Reviewer #1: The reviewed study entitled: “Difference in pyruvic acid metabolism between

neonatal and adult mouse lungs exposed to hyperoxia” by Tanaka et al. used CE TOFMS to

investigate the metabolic changes between hyperoxia in neonatal and adult mice. The analytical methods are sound and very well described in the manuscript. This is an interesting study and technique and the results will provide valuable information into using neonatal mice to investigate metabolic changes associated with respiratory exposures and disease.

Minor Comments:

1. In the final paragraph of the introduction on page three, the author should include some references on using CE TOFMS as an analytical tool for metabolomics. This would validate the technique and inform the readers on the how the technique was previously used.

2. Were any of the metabolites confirmed through MS/MS analysis combined with spectral library searching, or were they identified based on intact mass and electrophoresis? If they were only identified based on these 2 parameters, the author should include some information on the mass accuracy of the instrumentation used for the analysis and the mass error for the metabolite in parts per million (ppm).

6. PLOS authors have the option to publish the peer review history of their article (what does this mean?). If published, this will include your full peer review and any attached files.

Reviewer #1: No

---

## [Author Response · Author response to Decision Letter 0]

3 Aug 2020

Ch Ratnasekhar, Ph.D.

Academic Editor

PLOS ONE

1160 Battery Street, Koshland Building East, Suite 225

San Francisco, CA 94111, United States

Aug 03, 2020

Re: Manuscript ID PONE-D-20-17370 

Dear Dr. Ch Ratnasekhar, 

Please find attached a revised version of our manuscript entitled “Difference in pyruvic acid metabolism between neonatal and adult mouse lungs exposed to hyperoxia”, for publication as an Original Article in PLOS ONE.

Your comments and those of the reviewer were highly insightful and enabled us to greatly improve the quality of our manuscript. In the following pages are our point-by-point responses to each of the comments. Revisions in the text are shown using underline for additions, and strikethrough for deletions. We have deleted Ms. Kikumi Matsuoka’s name as an author and put her name in the Acknowledgement section.

We hope that the manuscript is now suitable for publication in PLOS ONE.

Yours sincerely, 

Fumihiko Namba, M.D., Ph.D.

Chief for Research

Center for Maternal, Fetal and Neonatal Medicine

Associate Professor of Pediatrics

Saitama Medical Center, Saitama Medical University

Journal Requirements:

1. Please ensure that your manuscript meets PLOS ONE’s style requirements, including those for file naming. The PLOS ONE style templates can be found at http://www.journals.plos.org/plosone/s/file?id=wjVg/PLOSOne_formatting_sample_main_body.pdf and http://www.journals.plos.org/plosone/s/file?id=ba62/PLOSOne_formatting_sample_title_authors_affiliations.pdf

We edited our manuscript according to PLOSOne formatting sample provided.

2. Please ensure you have thoroughly discussed any potential limitations of this study within the Discussion section, including the limited sample size.

We have discussed potential limitations of this study within the Discussion section.

3. PLOS ONE now requires that authors provide the original uncropped and unadjusted images underlying all blot or gel results reported in a submission’s figures or Supporting Information files. 

We have provided our original uncropped and unadjusted blot/gel image data are in Supporting Information.

Reviewer(s)' Comments to Author: 

Reviewer #1: The reviewed study entitled: “Difference in pyruvic acid metabolism between neonatal and adult mouse lungs exposed to hyperoxia” by Tanaka et al. used CE TOFMS to investigate the metabolic changes between hyperoxia in neonatal and adult mice. The analytical methods are sound and very well described in the manuscript. This is an interesting study and technique and the results will provide valuable information into using neonatal mice to investigate metabolic changes associated with respiratory exposures and disease.

Minor Comments:

1. In the final paragraph of the introduction on page three, the author should include some references on using CE TOFMS as an analytical tool for metabolomics. This would validate the technique and inform the readers on the how the technique was previously used.

As Reviewer recommended, references that reviews CE-TOFMS technique and that is using CE TOFMS are added.

2. Were any of the metabolites confirmed through MS/MS analysis combined with spectral library searching, or were they identified based on intact mass and electrophoresis? If they were only identified based on these 2 parameters, the author should include some information on the mass accuracy of the instrumentation used for the analysis and the mass error for the metabolite in parts per million (ppm).

We did not perform MS/MS. We identified metabolite based on intact mass and migration time in capillary electrophoresis. We assigned metabolite names by matching with the library created by measurement of reference material. Mass error was limited within ±10ppm of theoretical value and migration tome was limited within ±0.5 minutes of the library.

---

## [Editor Report · Decision Letter 1]

20 Aug 2020

Difference in pyruvic acid metabolism between neonatal and adult mouse lungs exposed to hyperoxia

PONE-D-20-17370R1

Dear Dr. Namba,

We’re pleased to inform you that your manuscript has been judged scientifically suitable for publication and will be formally accepted for publication once it meets all outstanding technical requirements.

Kind regards,

Ch Ratnasekhar, Ph.D.

Academic Editor

PLOS ONE
---

## [Editor Report · Acceptance letter]

24 Aug 2020

PONE-D-20-17370R1 

Difference in pyruvic acid metabolism between neonatal and adult mouse lungs exposed to hyperoxia 

Dear Dr. Namba:

I'm pleased to inform you that your manuscript has been deemed suitable for publication in PLOS ONE. Congratulations! Your manuscript is now with our production department. 

Kind regards, 

on behalf of

Dr. Ch Ratnasekhar 

Academic Editor

PLOS ONE